# Metabolic Activity of Human Embryos after Thawing Differs in Atmosphere with Different Oxygen Concentrations

**DOI:** 10.3390/jcm9082609

**Published:** 2020-08-12

**Authors:** Michal Ješeta, Andrea Celá, Jana Žáková, Aleš Mádr, Igor Crha, Zdeněk Glatz, Bartosz Kempisty, Pavel Ventruba

**Affiliations:** 1Department of Obstetrics and Gynecology, Faculty of Medicine, Masaryk University and University Hospital Brno, 602 00 Brno, Czech Republic; zakova.jana@fnbrno.cz (J.Ž.); crha.igor@fnbrno.cz (I.C.); ventruba.pavel@fnbrno.cz (P.V.); 2Department of Veterinary Sciences, Czech University of Life Sciences in Prague, 165 00 Prague, Czech Republic; 3Department of Biochemistry, Faculty of Science, Masaryk University, 602 00 Brno, Czech Republic; 323512@mail.muni.cz (A.C.); 175764@mail.muni.cz (A.M); glatz@chemi.muni.cz (Z.G.); 4Department of Nursing and Midwifery, Faculty of Medicine, Masaryk University, 602 00 Brno, Czech Republic; 5Department of Veterinary Surgery, Nicolaus Copernicus University, 87 100 Torun, Poland; bkempisty@ump.edu.pl; 6Department of Histology and Embryology, Poznan University of Medical Sciences, 61 701 Poznan, Poland; 7Department of Anatomy, Poznan University of Medical Sciences, 61 701 Poznan, Poland

**Keywords:** amino acids, human embryo, in vitro cultivation, capillary electrophoresis, fluorescence detection, metabolic activity

## Abstract

The vitrification of human embryos is more and more frequently being utilized as a method of assisted reproduction. For this technique, gentle treatment of the embryos after thawing is crucial. In this study, the balance of amino acids released to/consumed from the cultivation media surrounding the warmed embryos was observed in the context of a cultivation environment, which was with the atmospheric oxygen concentration ≈20% or with a regulated oxygen level—hysiological (5%). It is the first time that total amino acid turnover in human embryos after their freezing at post compaction stages has been evaluated. During this study, progressive embryos (developed to blastocyst stage) and stagnant embryos (without developmental progression) were analyzed. It was observed that the embryos cultivated in conditions of physiological oxygen levels (5% oxygen) showed a significantly lower consumption of amino acids from the cultivation media. Progressively developing embryos also had significantly lower total amino acid turnovers (consumption and production of amino acids) when cultured in conditions with physiological oxygen levels. Based on these results it seems that a cultivation environment with a reduced oxygen concentration decreases the risk of degenerative changes in the embryos after thawing. Therefore, the cultivation of thawed embryos in an environment with physiological oxygen levels may preclude embryonal stagnation, and can support the further development of human embryos after their thawing.

## 1. Introduction

The cryopreservation of human gametes, and mainly embryos, is a very important method in embryological laboratories worldwide. Vitrification is a very simple method, which does not require expensive programmable freezing equipment. The survival of cryopreserved oocytes and embryos is affected by many factors, and their effects on embryo vitality are still unclear. One of the most important factors limiting the success of cryopreservation is post-thaw embryo survival, affected by the formation of ice crystals during the freezing and thawing processes [1]. Such crystals may damage cell membranes and lead to blastomeric lysis [2]. Currently, there is no united procedure concerning how to treat human embryos after thawing. The warmed embryos are cultured just like non-cryopreserved fresh embryos. Fortunately, embryos are able to adapt to stress due to the variations in culture conditions. However, the cryopreservation imposes athe extra stress of the freeze–thaw process, which may affect cell homeostasis, metabolism, cell integrity and developmental potential. It has been documented in the literature that the warmed embryos are more sensitive and can be damaged by oxidative stress [3]. It has been described how oxidative stress is implicated in many different types of cell injuries, including cell membrane peroxidation, the oxidation of amino acids or nucleic acids fragmentation, associated with apoptosis and necrosis, which can dramatically decrease survival rate after cryopreservation [3,4]. An experiment on model animals indicated the positive effect of antioxidants present in the cultivation medium after the thawing of embryos [5].

A normal level of oxygen during in vivo oocyte and embryonic development is 2–8%. Actually, the embryos are exposed to a concentration gradient of oxygen during development. In follicles and the fallopian tube it is 5–8% [6], while in the uterus the oxygen concentration is lower, on the level of 1.5–2% [7]. The results of long term studies evidenced the positive effects of lower oxygen concentration (5%) on blastocyst rates and blastocyst cell number [8,9], implantation and pregnancy rates [10,11]. Indeed, the cultivation of precompacted embryos in the presence of 5% oxygen made a positive impact on the efficiency of fresh transfers [11], and also positively influenced implantation and pregnancy rates in the case of cryoembryostransfers [12].

The evaluation of metabolic activity in human embryos based on an analysis of cultivation media brings new information on their metabolic activity [13]. The evaluation of the developmental potential of human embryos according to the content of amino acids in the surrounding media is very complicated, and is associated with many unknowns. Nevertheless, a silence theory was established, [14] stating that human embryos with high developmental potential have lower metabolic demands, lower amino acid turnovers and have no extreme release or intake of amino acids from the media. This theory can be used for the evaluation of human embryos with total amino acid turnover as a parameter of embryo viability. This parameter has been evaluated as a potential prediction tool for the selection of quality embryos with a high implantation potential that correlates with pregnancy rates [15]. However, this theory was re-visited and it was concluded that embryos with maximal developmental potential are located in the “Goldilocks zone”. This means optimal metabolic level; “just the right” (not too high, not too low) metabolic activity [16]. Based on the total turnover of a small number of key amino acids, it is possible to distinguish between morphologically similar fresh or freezwe–thawed embryos [17], which have the capacity to form blastocysts [15], and embryos with a potential to achieve pregnancy [18]. Moreover, it has been presented that aneuploid embryos under in vitro conditions demonstrate different amino acid turnovers in comparison to their genetically normal counterparts [19].

Based on a previous study on embryo amino acid metabolism [15], the present study was designed to assess amino acid turnover (sum of production and consumption) during embryo development after vitrification.

In this study, total amino acid turnover was evaluated for the first time in human embryos after vitrification and cultivation in an environment with regulated—physiological—oxygen level (5%), or unregulated—atmospheric—oxygen concentration (≈20%). The objective of this work was to determine the effect of an environment with regulated oxygen level on the metabolome of human embryos after thawing. 

## 2. Experimental Section

All patients were under the age of 35 years, and they were treated according to standard protocols. Ovarian response was monitored by transvaginal ultrasonography and serum oestradiol. Oocyte collection was performed as previously described [20]. Oocytes were collected by follicular aspiration 36 h after hCG administration. The study was realized under the approval of the Ethics Committee of University Hospital Brno (Brno, Czech Republic). All patients participating in the study were informed by a responsible person and gave consent for their participation in this study. A total of 128 embryos from 128 women (mean age 33.5 years) were included in this study. After thawing and cultivation, the embryos were transferred to the uterus, with only one own embryo transferred to the uterus of one woman.

### 2.1. In Vitro Fertilization and Embryo Culture

Oocytes were fertilized in vitro by intracytoplasmic sperm injection—ICSI. The ICSI was performed according to the laboratory’s routine insemination procedures (day 0). On day 1, fertilization was checked 16–18 h after ICSI. The embryos were cultured in 400 µL of pre-equilibrated G TL™ medium (Vitrolife, Göteborg, Sweden) under a layer of Ovoil™ paraffin oil (Vitrolife, Göteborg, Sweden) in a 4-well NUNC dish in a Sanyo MCO-18M incubator at 37 °C, 5% CO^2^ and atmospheric oxygen concentration (≈20%). After 4 days of in vitro cultivation, selected embryos were transferred and all the other embryos suitable for cryoconservation were vitrified. The embryos were then vitrified individually in Rapid-i™ straws by using RapidVit™ Cleave (Vitrolife, Göteborg, Sweden). All steps were performed according to the manufacturer’s instructions. 

### 2.2. Cryoembryotransfer and Sample Collection

Randomly selected embryos were included in this study after patients’ agreement during the period 01/2017–12/2018. The embryos were warmed by RapidWarm™ Cleave (Vitrolife, Göteborg, Sweden) and placed into 25 µL drops on G-TL™ medium (Vitrolife, Göteborg, Sweden) covered by a layer of Ovoil™ paraffin oil (Vitrolife, Göteborg, Sweden) 24 h before planned cryoembryotransfer. 

For this study, only embryos frozen in the morula stage and cultured for 24 h after thawing were used. Selected embryos included in this study were cultivated in an atmosphere with 5.6% CO^2^, temperature 37 °C and atmospheric oxygen ≈20% (cultivation in Sanyo MCO-18M incubator), or in physiological atmosphere containing 5.6% CO^2^, 5% oxygen and 89.4% N^2^ (cultivation in Esco Miri TL incubator). After 24 h of cultivation one embryo was transferred to the uterus and the spent cultivation medium (G TL™, Vitrolife, Göteborg, Sweden) was placed individually into a PCR tube and stored in a freezer (−70 °C).

The embryos were assessed immediately after thawing by routine morphological criteria and subsequently cultured for 24 h. The development of the embryos after this cultivation was evaluated according to a grading system [21], and all the embryos were divided into groups of developing embryos (formed blastocoel and developed into blastocyst) and arrested embryos that were not able to form blastocyst within 24 h after thawing (Figure 1). A total of 61 embryos were included in the group with the atmospheric oxygen level; 41 (67.2%) of them showed progressive development after thawing and 20 (32.8%) were arrested. The group with the physiological oxygen level (5%) included 67 embryos, 47 (70.1%) with progressive development. All the embryos included in this study were transferred into a uterus after 24 h of in vitro cultivation. Clinical pregnancy was confirmed by ultrasound examination 4–6 weeks after embryo transfer by detection of the fetal heart beat.

### 2.3. Amino Acids Analyses

In total, samples of cultivation media collected from 128 embryos were analyzed. A total of 17 amino acids, Tau and Ala-Gln were determined using capillary electrophoresis with fluorescence detection. Sample preparation, derivatization and analysis of amino acids present in the media were done according to the procedure described in detail in a previously published paper [20]. In this procedure, 2 µL of spent cultivation media were mixed with 8 µL of acetonitrile for protein precipitation. The mixture was properly mixed and kept for 5 min at laboratory temperature in order to make a precipitate. The precipitate was then fractionated from the liquid using centrifugation for 10 min at 10,000× *g*. The sample was then derivatized with naphthalene-2,3-dicarboxaldehyde, accordingly. To the 5 µL of collected supernatant was added 15 µL of 8.33 mM sodium cyanide, 25 µL of the reaction buffer (200 mM boric acid/sodium hydroxide, pH 9.0) and finally 5 µL of 25 mM naphthalene-2,3-dicarboxaldehyde. The derivatization was conducted for 45 min at 25 °C and 650 rpm and the reaction was ended by moving the microtube with the reaction mixture into the freezer (70 °C). The derivatized samples were stored in a freezer and each sample was thawed just before analysis. Determination of the derivatized amino acids was made by the Agilent G7100A CE System (Agilent Technologies, Waldbronn, Germany) equipped with external collinear LEDIF detector ZETALIF™ LED 480 (Adelis, formerly Picometrics, Labège, France) (488/515 nm excitation/emission wavelength). Separation was performed in a fused silica capillary of 66.0 cm total length, 45.0 cm effective length and 50/375 µm inner/outer diameters. The samples were injected into the capillary at a pressure of 50 mbar for 3 s. The separation was conducted at 25 °C and the separation voltage was +30 kV. Baseline resolution of all the derivatized amino acids was accomplished in the background electrolyte composed of 50 mM sodium tetraborate, 73 mM sodium dodecyl sulphate, 5 mM sodium deoxycholate and 2.5 mM hydroxypropyl-β-cyclodextrin within 25 min. The concentrations of the amino acids present were determined from the parameters of calibration curves. The turnover of amino acids was quantified in pmol/embryo/hour. This was compared to the control samples in order to account for the changes in the composition of the medium which were not associated with embryonic growth.

### 2.4. Statistics

Results were expressed as amino acid consumption/production in pmol/embryo/hour. Each amino acid value was tested for significance from zero consumption/production using Student’s t-test. The differences between the amino acid profiles of embryos from the physiological and atmospheric oxygen levels were analyzed using ANOVA test (StatSoft, Inc. 2011, STATISTICA, version 10, Tulsa, OK, USA). Differences at *p* < 0.05 and *p* < 0.01 were considered statistically significant. The significance between proportions of clinical pregnancy and the groups was analyzed by Chi square test (*p* < 0.05).

## 3. Results

### 3.1. Development of Embryos after Thawing

The development of the embryos after thawing was very similar in both the observed groups. Of the 61 embryos cultured in the environment with an atmospheric oxygen level, 41 (67.2%) showed progressive development after thawing. The remaining 20 (32.8%) embryos were not able to develop into the blastocyst stage within 24 h after thawing and were labelled as arrested. The group cultured in the physiological oxygen level contained 67 embryos. Normal development to blastocyst after 24 h cultivation was observed in 47 (70.1%) of them. The other 20 (29.9%) could not form the blastocyst and stopped in the morula stage. These embryos were labelled as arrested. No statistically significant difference was found in the numbers of progressively developing embryos between the observed groups.

### 3.2. Total Amino Acid Consumption, Production and Turnover

Total turnover of amino acid production and consumption is a very good parameter for the evaluation of embryo viability, which has been documented several times [14,15,22]. The results of the present study revealed that the progressively developing embryos cultured in the environment with low oxygen concentration had a statistically (*p* < 0.05) lower amino acid turnover than the embryos that were cultured in the environment with the atmospheric oxygen concentration (Figure 2A). At the same time, it was found that the embryos cultured in the physiological oxygen environment made a significantly higher contribution of amino acids to the media in the group of those that were progressively developing when compared to the same embryos cultured in an atmospheric oxygen environment (Figure 2B). In addition, the cultivation of embryos in an environment with a regulated oxygen level is typically associated with a lower utilization of amino acid being taken in from the media, which was statistically significantly greater in the group cultured in a normal atmosphere for both the developing and the arrested embryos (Figure 2C).

### 3.3. Changes of Individual Amino Acids

Amino acids Leu, Ile, Phe, Met and Tyr were released into the culture medium in the atmosphere with a physiological level of oxygen containing the developing embryos (Figure 3A). These amino acids are present in the medium to high levels when compared to other amino acids [23]. They are important for early development and their significant production in conditions with 5% oxygen is interesting. For the arrested embryos cultured under atmospheric oxygen, there was a significant consumption of Tyr, His and Ile (Figure 3B). Furthermore, it was observed that the variation in amino acid utilization seemed greater when the embryos were cultured in atmospheric oxygen (Figure 3). It has been reported previously that embryos cultured in atmospheric oxygen manifested greater total amino acid utilization than embryos cultured in 5% of oxygen [24]. The results observed in the present study are very similar, and this can be caused by the consumption of amino acids. The embryos cultured in an atmospheric oxygen level exhibited more than a twofold increase in amino acid consumption in comparison to the embryos cultured in 5% of oxygen.

In the arrested embryos, a significantly higher production of amino acids into the cultivation media in the environment with 5% oxygen was also found in comparison to the developing embryos in the physiological atmosphere (Figure 2B). A comparison of the arrested and progressive embryos showed a significantly higher amino acid turnover in the arrested embryos in the environment with a physiologic oxygen level.

### 3.4. Development after Transfer

The clinical pregnancy rates after transfer of the embryos were statistically analyzed and no statistically significant differences were found between clinical pregnancy rates of embryos cultivated under atmospheric condition (22.9% pregnancy rate; 14 from 61 embryos) and embryos cultivated under physiological oxygen level conditions (23.8% pregnancy rate; 16 from 67 embryos).

## 4. Discussion

This study demonstrated that manipulation of embryos after thawing can make different changes in amino acid metabolism. These results are in agreement with previous studies, which demonstrated that 5% oxygen is more suitable for embryonic development when compared to the embryos cultured in an atmospheric oxygen level [24,25]. Our results correspond to past studies on the effects of 20% or 5% oxygen in human IVF, and support the current use of 5% oxygen [8]. It has been documented that human embryos after thawing were characterized by better development when cultured in the environment with 5% oxygen, compared to those cultured in the environment with 20% oxygen [6,7,8,9]. These embryos were frozen on the third day of cultivation and their development to the blastocyst stage was assessed. It was observed that the embryos in the environment with regulated oxygen level (5%) were characterized by faster growth, a lower apoptosis rate and a lower level of oxidative and heat shock stress [26]. The recent Cochrane review also showed that 5% oxygen concentration is related to increased live birth rate [10]. Therefore, it seems that oxygen level has a statistically significant effect on amino acid consumption from the cultivation media. Similar results were published for precompaction-stage human embryos [24].

When embryos are cultured in an oxygen concentration closer to the physiological state, they exhibit lower rates of amino acid turnover when compared to the embryos cultured in atmospheric oxygen (Figure 2A). The increased amino acid turnover in the embryos cultured in atmospheric oxygen can be attributed to a higher amino acid consumption. Similar to fresh embryos, in our results the thawed progressive embryos with the potential to develop to the blastocyst stage had a lower rate of amino acid consumption, production and total amino acid turnover than the arresting embryos with poor developmental potential.

It has been reported that changes in the amino acids in cultivation media during the development of human postcompaction embryos could be affected primarily by the concentration of amino acids in the cultivation medium [24]. The cultivation of the progressively developing embryos under the regulated atmosphere containing 5% oxygen was associated with an increase in the production of five amino acids (Leu, Ile, Phe, Met and Tyr). Interestingly, in the case of the arrested embryos, we also found a similar trend of higher amino acid contribution to the spent medium; however, we observed a significant increase in Tyr, Met and Ile, but not in Leu and Phe. As previously described, the embryos with a lower amino acid turnover are more viable. This is concordant with the earlier results, which show that mouse blastocysts with a lower glycolytic activity are more viable than embryos with high glycolytic metabolism [27]. Regarding differences between the developing and arrested embryos, this study did not confirm the earlier “silence hypothesis“, because the turnover of amino acids between the developing and arrested embryos was similar in normal atmosphere. However, in the case of the embryos from the physiological atmosphere, we found significantly lower amino acid turnover in the developing embryos in comparison to the arrested embryos. These results correspond with the silence theory.

It has been described that higher oxygen levels had a negative impact on mammalian embryos, especially on their viability [28]. Mammalian embryos during first the cleavage stages are very sensitive to oxygen damage [24]. Higher rates of ROS (reactive oxygen species), arising via oxidative embryonic metabolism, could be a major source of cellular stress during first days of embryonic development.

Total amino acid turnover correlates well with the viability and implantation ability of human embryos [29]. Wale and Gardner [24] found out that mouse embryos cultured in an environment with a low oxygen concentration expressed lower total amino acid turnover than mouse embryos cultured in an atmospheric environment.

Amino acid consumption and production during embryo development is a very interesting subject. It is known that oocytes and embryos have an endogenous reserve of all amino acids [30,31], which can therefore be released into the cultivation medium, but we cannot explain why this only happened in some dividing and in some arresting embryos.

It seems that the most successful embryos are characterized by a very quiet metabolic activity. They usually have usually lower requirements, and therefore lower amino acid turnover. The most vital embryos never have an extreme production or consumption of amino acids into/from the medium [14]. Manipulation with embryos after thawing in a physiological atmosphere is also associated with lower overall amino acid turnover. If the progressively developing embryos after thawing were cultured in conditions with lower oxygen concentrations, their amino acid turnover was significantly lower than if they were cultured in conditions with an atmospheric oxygen concentration. The process of regeneration after thawing is very important for the further development of embryos, and the present study demonstrated that the embryos cultured in a normal atmosphere had a statistically higher consumption of amino acids. The same trend was observed in mouse embryos during the first three days of cultivation, however an opposite trend was observed in 4–5-day-old embryos [32]. The experiments of the present study showed that both the progressively developing and the arrested embryos in the atmosphere with normal oxygen level are characterized by the increased consumption of amino acids when compared to those cultured in atmospheric oxygen (Figure 1). It seems that embryonic development under stress leads to the higher utilization of amino acids, and these data are in accordance with known facts [15]. The regeneration of embryos after thawing is a demanding process that involves the risk of damage by oxidative stress. In vitro studies also suggest that amino acids can act as antioxidants and buffers of intracellular pH in embryo cells [33].

For arresting embryos, a higher consumption of amino acids such as Leu, Ile and His is characteristic [14]. In the present study, significantly higher total consumption of Leu and Ile was observed in the progressively developing embryos cultured in the normal atmosphere. Statistically significant differences in the utilization of Leu, Ile, Phe and Met were found also in mouse embryos cultured in conditions with different oxygen concentrations. However, in this case, a higher consumption of these amino acids was recorded in the environment with regulated oxygen level [24]. Leu, Ile and Met were also presented as amino acids consumed by progressively developing human embryos [15]. Another study demonstrated that arrested warmed human embryos showed an increased consumption of Ile and Leu and, on the contrary, an increased production of Phe [17].

Met is an amino acid essential for embryonic development. The balance between Met and other amino acids is very important. Met has such an affinity for transporter molecules that if it is present in too high a concentration, it may disable the uptake of other amino acids. However, Met is required for the initiation of all protein syntheses through Met-tRNA [34]. The important protective function of Met residues against oxidative stress has been described previously. The oxidation of Met residues is reversible, and it works as a natural scavenging system against hazardous oxygen substances [35]. For this reason, Met can be consumed more than other amino acids in the presence of higher oxygen concentrations.

Leu was the amino acid most consistently depleted by embryos forming blastocysts throughout their development. This was an interesting finding; firstly, because Leu is an essential amino acid (see below), and secondly, it is an important signaling molecule involved in the stimulation of protein synthesis in many cell types [35,36,37]. Leu is also considered as an important amino acid for human embryo development, and it was found that its total turnover during the first days after insemination significantly correlated with clinical pregnancy and live birth rate [18]. Leu and Ile are very important amino acids for embryonal development, and they are contained in Vitrolife cultivation media (e.g., G2, which is very similar to the used GTL) in very high concentrations [23]. During our experiments, we observed the different reactions of embryos to cultivation in physiological or atmospheric oxygen levels. Finally, we observed a higher consumption of these amino acids, very important for embryonic development, mainly in progressive embryos. For both of these amino acids, we detected a significantly higher consumption by human embryos developing to blastocysts, compared to embryos arresting prior to the blastocyst stage. It was detected at day 2 and at the 8-cell stage of development. Moreover, Leu has been identified as the amino acid that is the most consistently used during the development of human embryos [15]. Similarly, Stokes et al. [17] presented significant differences between these two amino acids between the developed and arrested human embryos at day 2. During our experiments, we observed the different reaction of the embryos cultivated in either physiological or atmospheric oxygen levels. Finally, we observed, mainly in progressive embryos, the higher consumption of these amino acids in conditions with atmospheric oxygen.

Each metabolic profile can be associated with different oxidative stress conditions during thawing and cultivation. This can be supported by the significantly different levels of aromatic amino acids (Tyr and Phe) in medium between the two sample groups (5% and 20% oxygen). In the case of Phe, this was only present in the progressive embryos, whie Tyr was in both the arrested and progressive embryos. The production of Phe was also observed in a previous study, where a higher Phe production was detected in human developed embryos in comparison with arrested embryos at day 2 [17]. These amino acids have an antioxidant function, and are involved in the free radical scavenging activities of cells [38]. The presence of higher oxygen concentrations resulted in the higher consumption of this amino acid, similar to the case of Met. The exhausting of embryos after vitrification and the higher level of oxygen substances in medium probably resulted in the higher consumption of amino acids with scavenging ROS properties in human embryos after their thawing.

It is well known that Gln spontaneously breaks down to ammonium and pyroglutamate in a liquid medium; therefore, the media are supplemented with Gln in a dipeptide form, i.e., Ala Gln [27]. The loss of Ala Gln from the medium is greater with higher oxygen levels, although the release of Ala and Gln is lower. This means that both Ala and Gln are actively consumed for antioxidative processes. Gln supports the growth of cells that have high energy demands [39], whereas Ala can undergo transformation to pyruvate. Similarly, Glu, Asp and branched chain AAs can be transformed into α oxo acids with a potential antioxidant capacity [40]. Moreover, Glu is also a precursor of glutathione, which protects cells from oxidative damage [41].

It seems that embryos cultured under 20% oxygen exhibit the physiology of cells exposed to oxidative stress [41,42]. Under conditions with regulated levels of oxygen, the production of several essential amino acids was detected (Figure 3), while cultivation in conditions with high levels of oxygen was accompanied by their consumption. A similar effect was observed in mouse embryos where Ser, Glu and Asp were released to the medium in the atmosphere with 5% oxygen, but in unregulated atmospheric oxygen, these amino acids were consumed [43]. This fact can be associated with culture-induced stress as a result of atmospheric oxygen, which is more likely to induce the active metabolism and consumption of amino acids from medium. Higher overall consumption of amino acids was observed in both the developing and the arrested embryos only in the environment with the normal atmosphere.

In general, oxygen concentration in the uterus is very low (1.5–2%) [7], and so it seems that blastocysts need lower concentrations of oxygen than the other embryonic stages. A randomized study even revealed that when embryos were cultured in 5% oxygen on days 1–3, and then in 2% oxygen, their development was better than when the oxygen concentration was 5% throughout the cultivation [44].

## 5. Conclusions

In the developing and arrested embryos cultured in the physiological atmosphere, the consumption of amino acids was relatively low. From previous observations, it is well known that metabolically quiet embryos have better developmental competence. In this study, both the developing and the arrested embryos from the environment with regulated oxygen concentration (5%) were more metabolically quiet than the embryos from the normal atmosphere. These results indicate that the cultivation of human embryos after their thawing in a physiological oxygen concentration is less stressful and more convenient for human embryos.

## Figures and Tables

**Figure 1 jcm-09-02609-f001:**
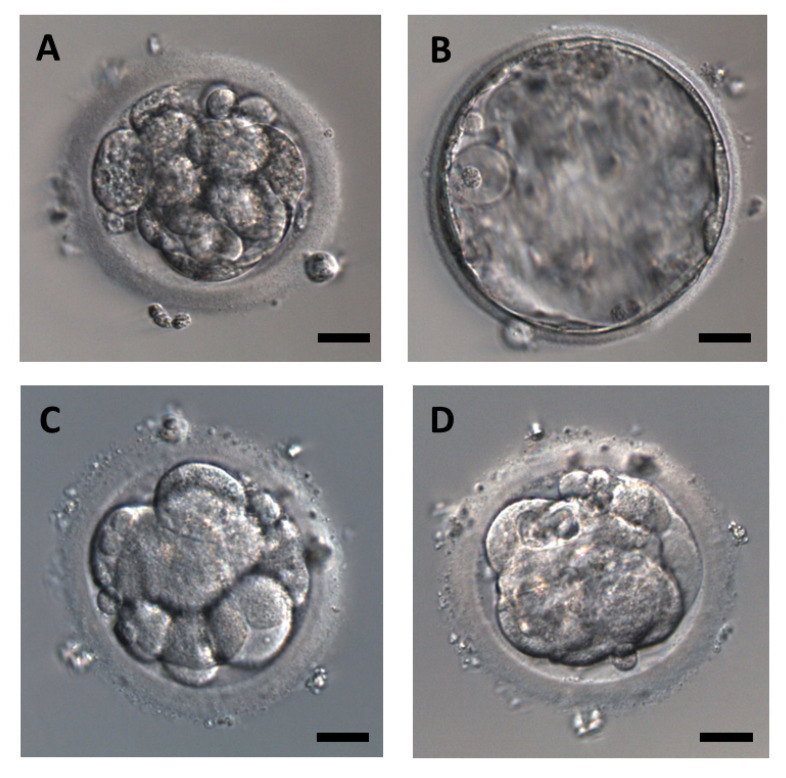
Representative images of human embryos after thawing (0 h, (**A**,**C**)) and after 24 h of cultivation (24 h, (**B**,**D**)) in progressive development (**A**,**B**) and arrested development (**C**,**D**). Scale bar represents 20 µm.

**Figure 2 jcm-09-02609-f002:**
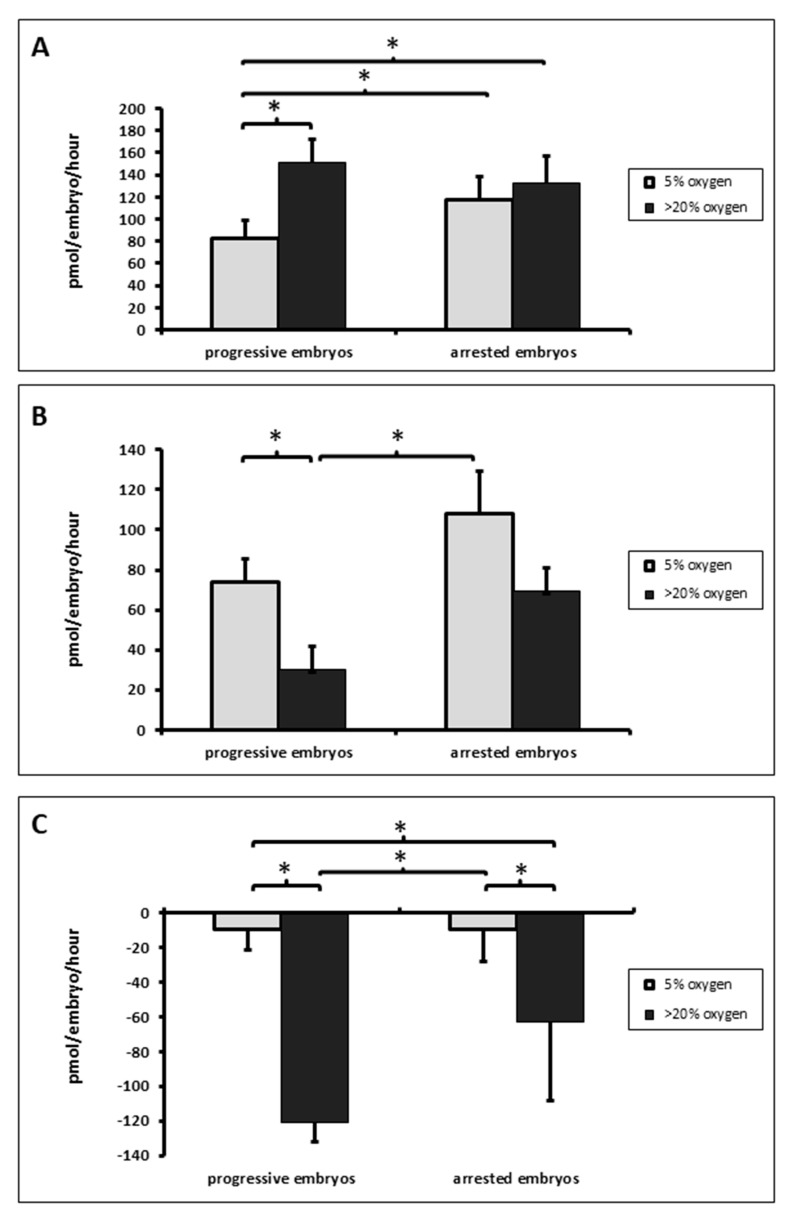
Amino acid turnover (**A**); production (**B**) and consumption (**C**) in progressively developing embryos which subsequently developed to the blastocyst stage, and embryos arrested prior to blastocyst formation. Values are the sum of the amino acids that significantly appeared/disappeared from the medium. Asterisks (*) show significantly different values (*p* ˂ 0.01).

**Figure 3 jcm-09-02609-f003:**
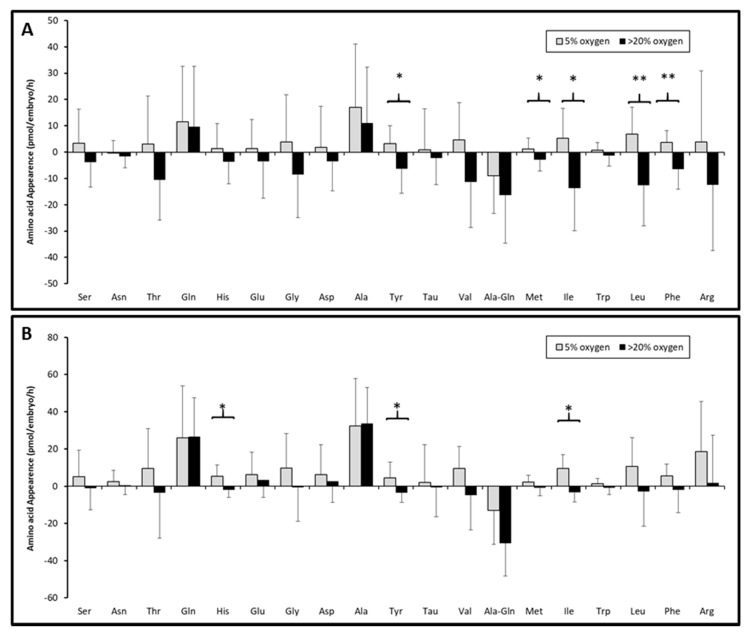
The bar chart showing the metabolic turnover of 17 amino acids, Tau and Ala-Gln in progressively developing embryos (**A**) and arrested embryos (**B**) cultured under the atmosphere containing either 20 oxygen (*n* = 61) or 5% oxygen (*n* = 67) for 24 h after vitrification. The error bars represent standard deviation. Asterisks show levels of statistical significance analyzed by ANOVA, *p* < 0.05; *, *p* < 0.01; **.

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
