# Peer review of "Metabolic Activity of Human Embryos after Thawing Differs in Atmosphere with Different Oxygen Concentrations"

_jcm, 2020, doi:10.3390/jcm9082609_

Round 1

Reviewer 1 Report

To authors,

The paper is well written and the data may contribute to better understanding of the human embryo physiology and also to improve the ART technology. I have some advice.

  1. Abstract should be more reader-friendly since not all readers are familiar with this issue. My concern regards “amino acid turnover” (core of this paper). You state here and there in the text, “total amino acid turnover as a parameter for embryo viability”, “Total turnover of amino acid production and consumption is a very good parameter for evaluation of embryo viability”, and “metabolically quiet embryos have better developmental competence”. Abstract does not reflect this context and thus many readers may be confused (as a specialist, I myself do understand the point). Please touch this meaning in Abstract, thereby making the context clearer.
  2. You state in the text, “In this study, total amino acid turnover was evaluated for the first time in human embryos after”, this should be much more emphasized also in the abstract and the text. Now that the competition among researches on this issue is very severe, you had better more straightforwardly state this “for the first-time issue”.
  3. You did not observe the real clinical/practical pregnancy success rate, right? Needless to say, the purpose/significance of this type of study is “clinical usefulness” and NOT “embryo-physiological understanding”. The latter is the basis of the former; however, considering that this journal is “clinical medicine” (not journal of reproductive biology), this standpoint is important. I mean the importance of “from the viewpoint of clinician”. Please consider to state (as a limitation) that clinical success/failure was not accessed in this study. Or such meaning. Very short statement is OK. Please do not expand the volume. The present manuscript is even too long.

Line 193. “Subsection”; typo?  

Author Response

Dear reviewer

Thanks for revision of our manuscript. Correction and changes are marked in suplement file. 

1) Abstract was revised and rewritten, we hope that now it will be more understandable

2) We agree with your recommendation, abstract was changed.

3) On behalf your note we add to manuscript another part regarding clinical pregnancy rate.

4) line 193: subsection was deleted

Reviewer 2 Report

There are missing files including tables and figures. Therefore, without figures and tables hard to give feedback on discussion.

Author Response

Dear reviewer

All results are added to uploading system. In text are now legends for each figure.

best regards

Reviewer 3 Report

In this study by Ješeta et. al. they address an important and current concern in culturing embryos at the proper oxygen level. While the percentage of developing vs. arresting embryos remain comparable at the two oxygen level they have examined, they found differences in amino acid turn over, potentially indicating sub-optimal conditions at 20% oxygen. This study adds to the number of published studies in the filed and is a valuable contribution. However, I have some concerns.

  • The English text in this manuscript is cumbersome and at times difficult to fully understand, specifically in the Methods and Results sections and needs major improvement.
  • The figures have no figure captions. And one has to infer the Figures from the text in the results section. Please add Figure captions.
  • The authors explain the potential role of the few amino acids that they find affected, but do not discuss why this aberrant amino acid turnover did not affect other amino acids and they should address this.
  • The authors have indicated that the embryos that were developing were transferred into the uterus. (They noted that 128 embryos were from 128 women. It is unclear if the relationship is one to one. Please clarify.) With regards to the developing embryos that were transferred, did the authors see any differences in live births based on the oxygen levels in this study? The authors do not mention any results in this regard.
  • The authors have used student T-test. These are multiple comparisons. They should use ANOVA together with multiple comparison correction for their statistical analysis.

Author Response

Dear reviewer

Thank you for revision of our manuscript. Changes are market in upload document.

Here are our notes: 1) english text in was revised especially in M/M

2) in text are new figure legends with description of figures

3) discussion was changed with new sentences about AA contents

4) In Results is new paragraph regarding pregnancy rate after transfer and section about transferrred embryos

5) Statistical evaluation of each amino acides changes was performed by ANOVA test and pregnancy rate was evaluated by Chi-square test. 

Round 2

Reviewer 2 Report

Thanks for sharing the detailed manuscript.
I think it can be accepted in the present form.
Thanks
Indrajit

Author Response

Thank you for your time with reading of our manuscript. In text we did small formal changes and correctionts of grammatical errors.

Reviewer 3 Report

1. Figure 2 and its legend: the use of letters, A/B to determine statistical significance is confusing. Use brackets between the comparisons AND indicate which comparisons are statistically significant, as was done in Fig 3.

2. In Figure 3, the error bars indicate standard error of the mean (SEM). This is incorrect. The authors should show the standard deviation (SD) instead. SEM is smaller than SD, because it shows the deviation of the sample mean from the population mean, which is not analyzed in this study. Rather the mean was calculated and SD would show deviation from the mean.

3. The authors have edited the discussion, but have not addressed a previous comment regarding why their observed aberrant amino acid turnover affected some but not other amino acids. They should offer some explanations together with examples from other studies.

4. The text has improved greatly, but there are still grammatical errors throughout the text.

Author Response

Thank you for revision of our manuscript. In reaction on your comments we did this changes in manuscript:

  1. Figure 2 is comletely changed and now is with asterics and brackets. page 8
  2. In Figure 3 error bars was changed and now show standard deviation (SD) page 10
  3. In discussion we add several sentences regarding impact of lack amino acids on other amino acids.                                                                           page 12 line 323-327,  334-347  page 13 352-358
  4. Text was completely revised and it was corrected several grammatical errors.
